

# Vertical variability of aerosol properties and trace gases over a remote marine region: A case study over Bermuda

Taiwo Ajayi[1], Yonghoon Choi[2], Ewan C. Crosbie[2,3], Joshua P. DiGangi[2], Glenn S. Diskin[2], Marta A. Fenn[2,3], Richard A. Ferrare[2], Johnathan W. Hair[2], Miguel Ricardo A. Hilario[1], Chris A. Hostetler[2], Simon Kirschler[4], Richard H. Moore[2], Taylor J. Shingler[2], Michael A. Shook[2], Cassidy Soloff[1], Kenneth L. Thornhill[2], Christiane Voigt[4,5], Edward L. Winstead[2,3], Luke D. Ziemba[2], Armin Sorooshian[1,6]

[1]Department of Hydrology and Atmospheric Sciences, University of Arizona, Tucson, AZ, 85721, USA
[2]NASA Langley Research Center, Hampton, VA, 23681, USA
[3]Analytical Mechanics Associates, Hampton, VA, 23666, USA
[4]Institute of Atmospheric Physics, German Aerospace Center, Germany
[5]Institute of Atmospheric Physics, University Mainz, Germany
[6]Department of Chemical and Environmental Engineering, University of Arizona, Tucson, AZ, 85721, USA

*Corresponding author: armin@arizona.edu





**Abstract.** Remote marine regions comprise a high fraction of Earth's surface, but in-situ vertically-resolved

measurements over these locations remain scarce. Here we use airborne data during 15 vertical spiral soundings (0.15 – 8.5 km) over Bermuda during the NASA Aerosol Cloud meTeorology Interactions over the western ATlantic Experiment (ACTIVATE) to investigate the impact of different source regions on the vertical structure of trace gases, aerosol particles, and meteorological variables over 1000 km offshore of the U.S. East Coast. Results reveal significant differences in vertical profiles of variables between three different air mass source categories (North America, Ocean,

Caribbean/North Africa) identified using the HYSPLIT model: (i) the strongest/weakest pollution signature from North America/Ocean category; (ii) North American air has the highest levels of CO, $CH_4$, submicron particle number concentration, AMS mass, and highest organic mass fraction along with smoke layers in free troposphere (FT); (iii) Ocean air has the highest relative amount of nitrate, non-sea-salt sulfate, and oxalate, which are key acidic species participating in chloride depletion; (iv) pronounced coarse aerosol signature in the FT and reduced aerosol

hygroscopicity in air masses from the Caribbean/North Africa associated with dust transport; and (v) considerable vertical heterogeneity for almost all variables examined, including higher $O_3$ and submicron particle concentrations with altitude, suggestive that the FT is a potential contributor of both constituents in the marine boundary layer. This study highlights the importance of considering air mass source origin and vertical resolution to capture aerosol and trace gas properties over remote marine areas.






## 1 Introduction

Atmospheric studies are important to conduct over oceans, which represent a large fraction of Earth's surface area. Remote marine areas are particularly important because sometimes they represent cleaner conditions with lower aerosol number concentrations, and clouds in such clean conditions account for much of the diversity in observational and model calculations for radiative forcing from aerosol–cloud interactions (Gryspeerdt et al., 2023). While it may seem reasonable to assume that such clean conditions are more attainable far away from continents (> 1000 km), such

remote areas may not be clean due to long-range transport. Even if the atmosphere is not pristine at such offshore distances, characterizing such environments is helpful for comparisons to reanalysis, remote sensing, and model output data. In particular, vertically-resolved in-situ data for aerosol properties and trace gases are difficult to obtain over remote marine areas, which presents challenges for validation and improvement of models and reanalysis products (Easter et al., 2004; Edwards et al., 2022; Fernandes et al., 2023) since past work has shown significant deviations in

aerosol properties at the surface versus higher levels more relevant for clouds (Pringle et al., 2010). Such data from airborne platforms are necessary to better determine how the vertical distribution of particles and trace gases impact atmospheric processes such as cloud life cycle and radiative transfer.

The island of Bermuda in the northwest Atlantic is more than 1000 km removed from the U.S. East Coast and receives transported emissions from multiple continents such as North and Central America, Asia, Europe, and Africa (Holben

et al., 2001; Smirnov et al., 2002). This blend of seasonally dependent continental sources and marine emissions has drawn researchers to Bermuda and the adjacent Sargasso Sea for decades (e.g., Wolff et al., 1986; Galloway et al., 1989; Moody et al., 1995; Anderson et al., 1996; Arimoto et al., 2003; Keene et al., 2014; Aldhaif et al., 2021). Table S1 summarizes the rich body of past studies conducted at Bermuda. While much is known about the surface characteristics of trace gases and aerosol particles at Bermuda, much less is known aloft with the exception of a few

studies which compare aerosol size distribution characteristics in the boundary layer and free troposphere, showing higher number/volume concentrations (and associated mean diameters of number/volume) at lower altitudes (Horvath et al., 1990; Kim et al., 1990). There are knowledge gaps for other aerosol properties (e.g., new particle formation, aerosol composition, hygroscopicity, and optical properties) and trace gases, especially as it relates to their vertical distribution and sensitivity to air mass source region. Aryal et al. (2014) showed that surface optical data were weakly

related to column optical properties over Bermuda, strongly motivating more meticulous examination above the surface where there can be heterogeneity in gas and aerosol characteristics.

The goal of this work is to use a series of aircraft vertical spiral soundings conducted around Bermuda during the NASA ACTIVATE airborne mission to characterize the vertical distribution of aerosol particle characteristics and trace gas levels. We also examine differences in the measurement data between the boundary layer and free

troposphere as a function of air mass source. Section 2 summarizes the ACTIVATE field campaign measurements used in this work along with calculation methods. Section 3 presents the results and discussion. Section 4 provides conclusions.





## 2 Methods

### 2.1 Flight campaign

The NASA Aerosol Cloud meTeorology Interactions oVer the western ATlantic Experiment (ACTIVATE) involved three years of flights with separate deployments each year in winter and summer months (Sorooshian et al., 2019, 2023). ACTIVATE involved two NASA Langley Research Center (LaRC) aircraft flying in spatial coordination. A HU-25 Falcon flew in and above the marine boundary layer (usually < 3 km) while a King Air flew around ~9 km. The Falcon conducted in-situ measurements of trace gases, aerosol particles, meteorological variables, and clouds

while the King Air carried out remote sensing and launched dropsondes. Flights were mostly based out of NASA LaRC in Hampton, Virginia with durations of <4 hours. Due to restrictions associated with the COVID-19 pandemic, the possibility of flying to Bermuda only opened in the final deployment (Summer 2022), prompting the mission team to conduct an intensive operation based at L.F. Wade International Airport (Bermuda) between 31 May and 18 June (2022); this period included research flights 161-179 with the first and last in this group being transit flights back and

forth from Virginia to Bermuda. The objective in the current study is a detailed examination of data collected from a total of 15 vertical spiral soundings (~0.15  to as high as ~8 km) with the HU-25 Falcon during the Bermuda intensive period (details in Table 1 and locations in Fig. 1). Most of the Falcon spirals were coordinated with the King Air such that the latter would fly over the Falcon's spiral location to provide vertically-resolved profiles of aerosol properties.






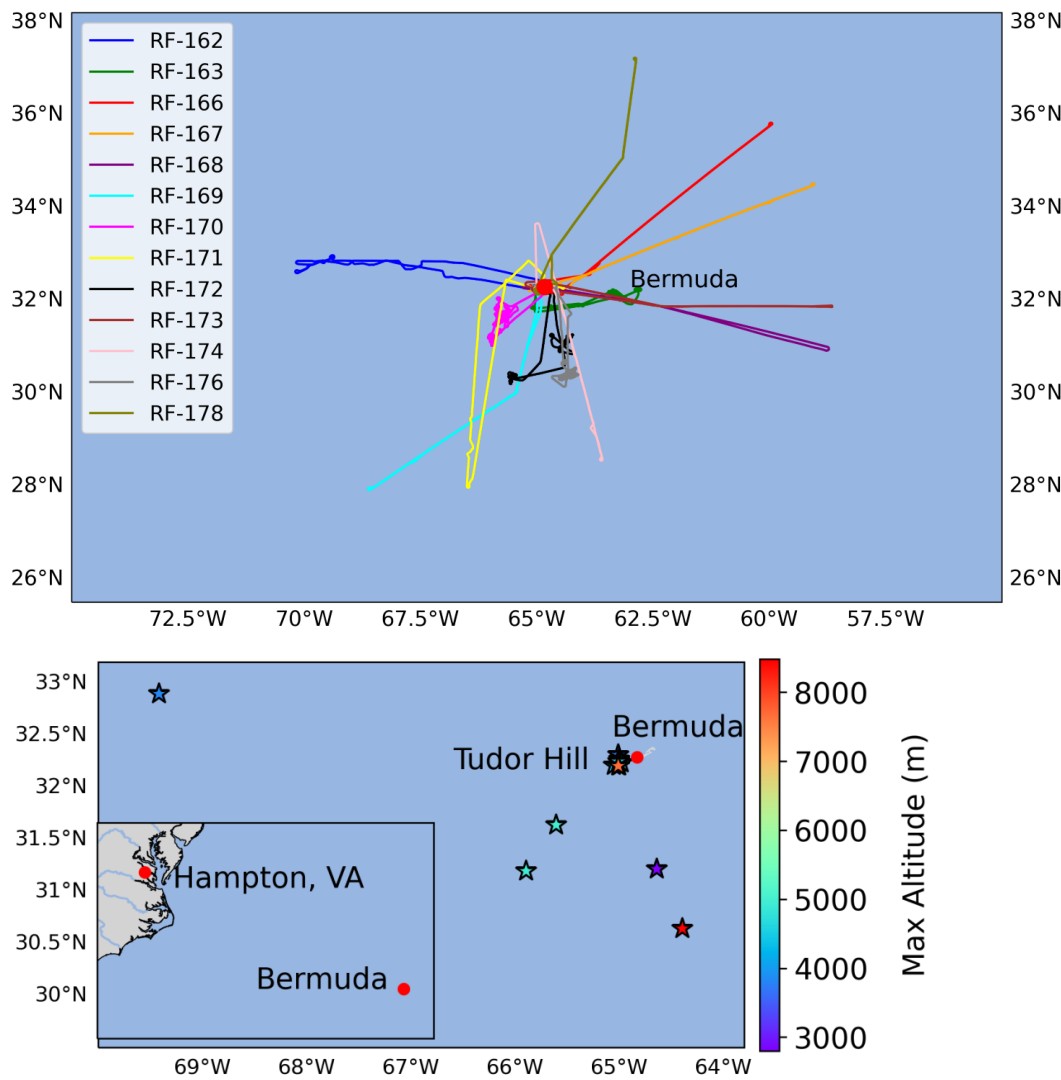

Figure 1: (Top) Flight tracks corresponding to the 13 flights that included the 15 Falcon spiral soundings, colored by research flight (RF) number. Bermuda is marked with a red marker. (Bottom) Location of the Falcon spiral soundings colored by the maximum altitude reached by the Falcon. The inset map shows the location of Bermuda relative to North America and the main base of operations during ACTIVATE (NASA Langley Research Center in Hampton, Virginia). Tudor Hill is labeled for context since it is an extensively used research site by the southwestern tip of Bermuda that was close to the location of most spirals.


**Table 1.** Summary of details relevant to the 15 Falcon spiral soundings used in this study. APT means accumulated

precipitation along trajectories ending at the location of the Falcon (described in Sect. 2.3). MBLH = calculated marine



boundary layer height from Falcon data (Sect. 2.4.2); MLH = HSRL-2 derived values of aerosol mixed layer height (NA means no data were available during the portion of the flight over the location of the Falcon's spiral sounding).

| Research flight - date | Spiral stop and start time (UTC) | Altitude range (m) | Falcon MBLH/HSRL-2 MLH (m) | APT (mm) | Air mass source category |
|---|---|---|---|---|---|
| 162 - 06/02/2022 | 12:58 – 13:10 | 150 - 4000 | 1050/NA | 3.23 | Ocean |
| 162 - 06/02/2022 | 13:48 – 14:10 | 150 - 3200 | 1050/NA | 2.67 | Ocean |
| 163 - 06/02/2022 | 18:36 – 18:53 | 150 - 4000 | 750/NA | 2.18 | Ocean |
| 166 - 06/07/2022 | 14:32 – 14:47 | 150 - 3200 | 950/945 | 0.67 | North America |
| 167 - 06/07/2022 | 19:08 – 19:23 | 150 - 3500 | 1200/1100 | 0.45 | North America |
| 168 - 06/08/2022 | 20:30 – 20:45 | 150 - 4000 | 1250/1100 | 0.07 | North America |
| 169 - 06/10/2022 | 15:17 – 15:25 | 150 - 3500 | 800/NA | 7.14 | Caribbean/North Africa |
| 169 - 06/10/2022 | 19:30 – 19:51 | 150 - 5000 | 800/700 | 7.60 | Caribbean/North Africa |
| 170 - 06/10/2022 | 20:25 – 20:35 | 150 - 5000 | 1190/NA | 8.73 | Caribbean/North Africa |
| 171 - 06/11/2022 | 15:24 – 15:42 | 150 - 5000 | 1190/NA | 6.16 | Caribbean/North Africa |
| 172 - 06/11/2022 | 20:08 – 20:30 | 150 - 3000 | 1350/1110 | 6.14 | Caribbean/North Africa |
| 173 - 06/13/2022 | 14:30 – 14:46 | 150 - 4500 | 805/815 | 2.12 | Caribbean/North Africa |
| 174 - 06/13/2022 | 19:46 – 20:09 | 150 - 4500 | 915/940 | 2.65 | Caribbean/North Africa |
| 176 - 06/14/2022 | 16:05 – 16:25 | 150 - 8500 | 1210/1240 | 3.14 | Caribbean/North Africa |
| 178 - 06/17/2022 | 16:20 – 16:47 | 150 - 7600 | 1210/1230 | 9.67 | North America |

## 2.2 Instrumentation

Table 2 summarizes instrumentation details relevant to this study on both the Falcon and King Air with a brief description provided below. Extensive details regarding all ACTIVATE flight and instrumentation are provided elsewhere (Sorooshian et al., 2023). Beginning with in-situ measurements on the Falcon, a PICARRO G2401-m instrument measured CO, $CO_2$, and $CH_4$, while the 2B Tech. Inc. Model 205 instrument measured $O_3$. The Picarro G2401-m sampled air via an air temperature probe (Buck Research Instruments, LLC) on the aircraft's crown next to

the aerosol inlets. Calibrations were conducted hourly during flights and weekly on the ground. Ozone measurements relied on using a forward-facing J-probe inlet on the HU-25 Falcon nadir panel with a specialized sampling device to increase data quality at high altitudes. Data quality checks were conducted hourly during flight such as zeroing for 1 min using a KI filter. Water vapor measurements were conducted with a diode laser hygrometer (DLH), which is an open path, near-infrared absorption spectrometer. Its optical path was outside the aircraft between an aircraft window

and a retroreflector on a starboard wing pylon.

Aerosol number concentrations in different size ranges were measured with multiple condensation particle counters, and size distributions extending from 3 to 5000 nm were obtained with a scanning mobility particle sizer (SMPS; 3 – 100 nm) and laser aerosol spectrometer (LAS; 100 – 5000 nm). The SMPS categorizes particles according to their electrical mobility diameters, while LAS categorizes them based on their optical diameters (Sorooshian et al., 2023).



The LAS is calibrated using monodisperse ammonium sulfate particles since they have a refractive index (1.52) representative of ambient aerosol particles (Shingler et al., 2016; Aldhaif et al., 2018). The archived SMPS and LAS data are adjusted so the SMPS electric mobility diameters matches the optical diameters of the LAS, allowing size distributions from the two instruments to be stitched together (Sect. 3.5, Sorooshian et al., 2023). The stitched size distributions had fits made to them with a non-linear least squares fit of a two-mode lognormal distribution using the

Python package scipy.optimize.curve_fit (Virtanen et al., 2020).

The hygroscopicity parameter f(RH) for submicron aerosol was measured using tandem nephelometers with relative humidities set to approximately 20% and 80% (Ziemba et al., 2013). Humidification was conducted with a custom-made humidifier, and the f(RH) was calculated as the total light scattering (550 nm) at 80% relative to 20%. While optical properties were characterized (e.g., scattering, absorption), those results are the subject of separate work related

to the larger dataset from flights during June 2022. The high-resolution time-of-flight aerosol mass spectrometer measured submicron non-refractory aerosol chemical composition (DeCarlo et al., 2008), including mass concentrations of sulfate, nitrate, chloride, ammonium, and organics with a vacuum aerodynamic diameter window from 60 to 600 nm. Data were collected using a pressure-controlled inlet (500 Torr) and averaged to 30 s time resolution using the native 1 Hz fast-MS-mode measurements. The instrument's ionization efficiency was calibrated

with 400 nm ammonium nitrate particles. A collection efficiency of one was applied to the data after comparison to sulfate measured by the PILS instrument, which is described subsequently. This study makes use of the spectral marker m/z 44, which signifies oxygenated organic species (Zhang et al., 2005). The ratio of m/z 44 to total organics is referred to as $f_{44}$ and of interest in this study as work from ACTIVATE (Dadashazar et al., 2022) and other regions (Sorooshian et al., 2010) show an enhancement in this ratio associated with cloud processing. To allow for better data quality, m/z

44 data are used when total AMS mass concentration exceeded 0.4 μg m$^{-3}$.

Water-soluble aerosol composition data were collected with a particle-into-liquid sampler (PILS) coupled to offline ion chromatography (Crosbie et al., 2022). The PILS-IC data are at 5 min resolution and serve to complement the AMS data by allowing for a look into aerosol composition up to a diameter of 5 μm, in addition to speciating tracers for sea salt (e.g., Na$^+$) and dust (i.e., Ca$^{2+}$), which is not possible with the AMS. We caution that denuders were not

used and thus some species such as nitrate may be subject to artifacts. Details of the IC analysis are provided in detail elsewhere (Gonzalez et al., 2022; Corral et al., 2022b). Non sea salt sulfate (nss SO$_4^{2-}$) from the PILS-IC was calculated using the ratio of sulfate to sodium in pure sea water assuming all sodium came from sea salt. For help with cloud screening to avoid droplet shatter artifacts, data are only used when the wing-mounted fast cloud droplet probe (FCDP; 3-50 μm diameter) measured a liquid water content (LWC) below 0.02 g m$^{-3}$ and cloud droplet number concentration

less than 10 cm$^{-3}$ (Kirschler et al., 2023). The FCDP is a forward-scattering probe on the port wing of the Falcon with extensive processing and corrections applied as summarized by Kirschler et al. (2022). Only two of the 15 spirals included a cloud layer (RF 168 on 8 June 2022 and RF 178 on 17 June 2022), which was confirmed using the Falcon forward camera.

The higher-flying King Air obtained vertical profiles of temperature, relative humidity, and wind speed via dropsondes

launched with the National Center for Atmospheric Research (NCAR) Airborne Vertical Atmospheric Profiling System (AVAPS) (Vömel et al., 2023). Dropsondes were strategically positioned within reasonable proximity to the



location of the Falcon spiral soundings to offer a meaningful intercomparison. The data span from approximately 9 km to the surface, with the former being the typical flight altitude of the King Air. We use vertically-resolved data of aerosol extinction, aerosol depolarization ratio, and lidar ratio as measured by the second generation High Spectral Resolution Lidar (HSRL-2) instrument (Hair et al., 2008; Burton et al., 2018). This instrument provides vertically-resolved values of aerosol backscatter and depolarization at 355, 532, and 1064 nm along with independent measurements of aerosol extinction at 355 and 532 nm using the HSRL method. Additionally we use the aerosol mixed layer height (MLH) (Scarino et al., 2014) and aerosol type (Burton et al., 2012) products derived from the HSRL-2 base products. The King Air did not fly on 2 June 2022 owing to maintenance requirements and thus did not provide data for the Ocean category, which coincidentally involved flights (RF 162-163) on just that one day.

**Table 2.** Summary of HU-25 Falcon and King air instrumentation from which data are used in this study.

| Observable | Instrument | Diameter Range (μm) | Time resolution (s) | Reference |
|---|---|---|---|---|
| | | HU-25 Falcon | | |
| Aerosol Number Concentration | TSI-3776 condensation particle counter (CPC) | 0.003 – 5 | 1 | Xiao et al. (2023) |
| Aerosol Number Concentration | TSI-3772 CPC | 0.01 – 5 | 1 | Xiao et al. (2023) |
| Dry Aerosol Size Distribution | TSI scanning mobility particle sizer (SMPS): Model 3085 differential mobility analyzer (DMA) Model 3776 CPC and Model 3088 neutralizer | 0.003 – 0.1 | 45 | Xiao et al. (2023) |
| Dry Aerosol Size Distribution | TSI 3340 laser aerosol spectrometer (LAS) | 0.1 - 5 | 1 | Moore et al. (2021) |
| f(RH) Hygroscopicity | TSI-3563 nephelometer with 80 % humidification | <1 | | Ziemba et al. (2013) |
| Speciated Non-refractory Mass ($SO_4^{2-}$, $NO_3^-$, $NH_4^+$, $Cl^-$, Org, m/z 44) | Aerodyne High Resolution Time of Flight Aerosol Mass Spectrometer (HR-ToF-AMS) | 0.06 – 0.6 | 25 | DeCarlo et al. (2008) |
| Speciated Water-Soluble Aerosol Mass | BMI Particle-Into-Liquid Sampler (PILS) Coupled to Offline Ion Chromatography | < 5 μm | 300 | (Crosbie et al., 2022) |
| Relative Humidity Water Vapor | Diode Laser Hygrometer | n/a | 1 | Diskin et al. (2002) |
| $CH_4$, CO, $CO_2$ | PICARRO G2401-m | n/a | 2.5 | DiGangi et al. (2021) |
| $O_3$ | 2B Tech. Inc. Model 205 | n/a | 2 | Wei et al. (2021) |
| Liquid Water Content and Drop (LWC) Concentration ($N_d$) | Fast Cloud Droplet Probe (FCDP) | 3 - 50 | 1 | Kirschler et al. (2022) |
| GPS Altitude, Longitude, Latitude | Applanix 610 (navigational) | n/a | 1 | Thornhill et al. (2003) |



| | | | | |
|---|---|---|---|---|
| Temperature | Rosemount 102 sensor | n/a | 0.05 | |
| King Air | | | | |
| Temperature, relative humidity, wind speed | Vaisala NRD41 dropsonde (meteorological state) | n/a | | Vömel et al. (2023) |
| Aerosol depolarization ratio (532 nm), mixed layer height (MLH) | High spectral resolution lidar (HSRL-2) | n/a | 10 | Hair et al. (2008); Scarino et al. (2014); Burton et al. (2015) |
| Aerosol extinction, lidar ratio (532 nm), aerosol type | High spectral resolution lidar (HSRL-2) | n/a | 60 | Hair et al. (2008); Burton et al. (2012) |
| GPS Altitude, Longitude, Latitude | Applanix 610 (navigational) | n/a | 1 | Thornhill et al. (2003) |

### 2.3 Air mass source identification

A critical aspect of this study was to identify the source of air arriving at different vertical levels of each spiral sounding. To do this, 72 hour back-trajectories were obtained using the NOAA Hybrid Single-Particle Lagrangian Integrated Trajectory model (HYSPLIT) (Stein et al., 2015; Rolph et al., 2017) arriving at different vertical levels (1 minute apart) of each spiral. Although not shown, 10 day back-trajectories were also used to corroborate air mass type assignments. Trajectories were produced using the Global Data Assimilation System (GDAS, 1° x 1° global resolution) archive data and the "model vertical velocity" approach. Additionally, accumulated precipitation along trajectories (APT) was estimated by summing up the precipitation at each one-hour time step along each trajectory to the spiral sounding location. APT is a proxy variable to represent the possibility of aerosol removal via wet scavenging along trajectory paths (Dadashazar et al., 2021; Hilario et al., 2021).

### 2.4 Calculations

#### 2.4.1 Air mass source categories

HYSPLIT trajectories were divided into three categories based on the source region each trajectory spent the most time in before arriving at a distinct vertical level of individual spiral soundings. The three source regions include Caribbean/North Africa, North America, and Ocean, with the lattermost being primarily confined to the region surrounding Bermuda. Note that North African parcels generally curve around the Caribbean region (and thus combined as one category) with possible influence also from southeast U.S. on their path towards Bermuda. This is a common summertime trajectory pattern over the northwest Atlantic (Sorooshian et al., 2020; Painemal et al., 2021). Trajectory paths ending at vertical levels in and above the boundary layer were similar for each spiral sounding and thus assignments of air mass source origin apply to the entire column of individual spirals (Table 1).




### 2.4.2 Altitude categories for data analysis

We determined the MBL height (MBLH) for each spiral sounding using a well-established approach relying on the maximum vertical gradient of virtual potential temperature, specific humidity, and aerosol concentration (Seibert et al., 2000), as measured by the Falcon. Values range from 750 to 1350 m (Table 1), which are mostly similar to the

HSRL-2 values of aerosol MLH that are independently derived using steep gradients in aerosol backscatter profiles (Scarino et al., 2014). To allow for easier comparison of data between spirals, a subset of results separates data into those collected below 1 km and those between 1 and 3.5 km as a way to simplify a look into the marine boundary layer (MBL) and free troposphere (FT), while avoiding bias in results for spirals ending at much higher maximum altitudes than others. Although three spirals only reached 3-3.2 km (1 in each air mass category), we use 3.5 km to

increase the range of altitudes into the FT (i.e., more statistics) for the majority of spirals. For a more complete view of vertical profiles, some figures additionally show the full altitude range of relevant measured variables.

### 3. Results

### 3.1 Air mass sources and weather characteristics

The back-trajectory results for the 15 aircraft spiral soundings reveal that the three predominant categories are Ocean (Fig. 2a-c), North America (Fig. 2d-g), and Caribbean/North Africa (Fig. 2h-o). The Ocean category includes trajectories more closely confined to the marine areas to the west of Bermuda. The North America category has trajectories coming from North America, with the case on 17 June having some mixing with North Atlantic air. The range of transport times from the coastline of North America to Bermuda for this category was ~2-3 days hours for

the MBL and ~2 days for the FT. The third category has air masses coming from the southwest by the Caribbean region, with many trajectories derived from North Africa (confirmed with 10-day back-trajectories).

Ocean trajectories are presumed to have more influence from marine-derived emissions such as sea spray and products of marine biogenic emissions of species such as dimethylsulfide (DMS) (Luria et al., 1989; Andreae et al., 2003). North American air masses commonly have more influence from the wide range of anthropogenic sources common

to the eastern U.S. including urban emissions (e.g., vehicular, industrial, combustion), agricultural emissions, and biomass burning (Corral et al., 2021). Finally, the Caribbean/North Africa category is suspected to have more influence from dust, biomass burning, and agricultural emissions, the latter two of which are common around the southeast U.S (Sevimoglu and Rogge, 2015; Le Blond et al., 2017; Corral et al., 2020; Edwards et al., 2021; Mardi et al., 2021) and the dust derived often from North Africa (Arimoto et al., 1995; Anderson et al., 1996; Todd et al., 2003; Muhs et al.,

235 2012).

Regarding influential weather during the time frame of this study, 5-6 June 2022 was marked by the arrival of tropical storm Alex, which transitioned to an extratropical storm and left to the northeast of Bermuda prior to the 7 June flight. Starting 7 June, a high pressure system moved into the Bermuda area through 9 June, which was a period marked by the arrival of North American air. After that point, cold fronts moving off the U.S. East Coast shifted the high pressure





system farther east setting up the classical sub-tropical Bermuda-Azores High that promotes southwesterly flow into the area, which explains the string of Caribbean/North Africa air mass days between 10-14 June. Corresponding to the last spiral on 17 June with North American air, there was another high pressure system that set up over the Bermuda area.

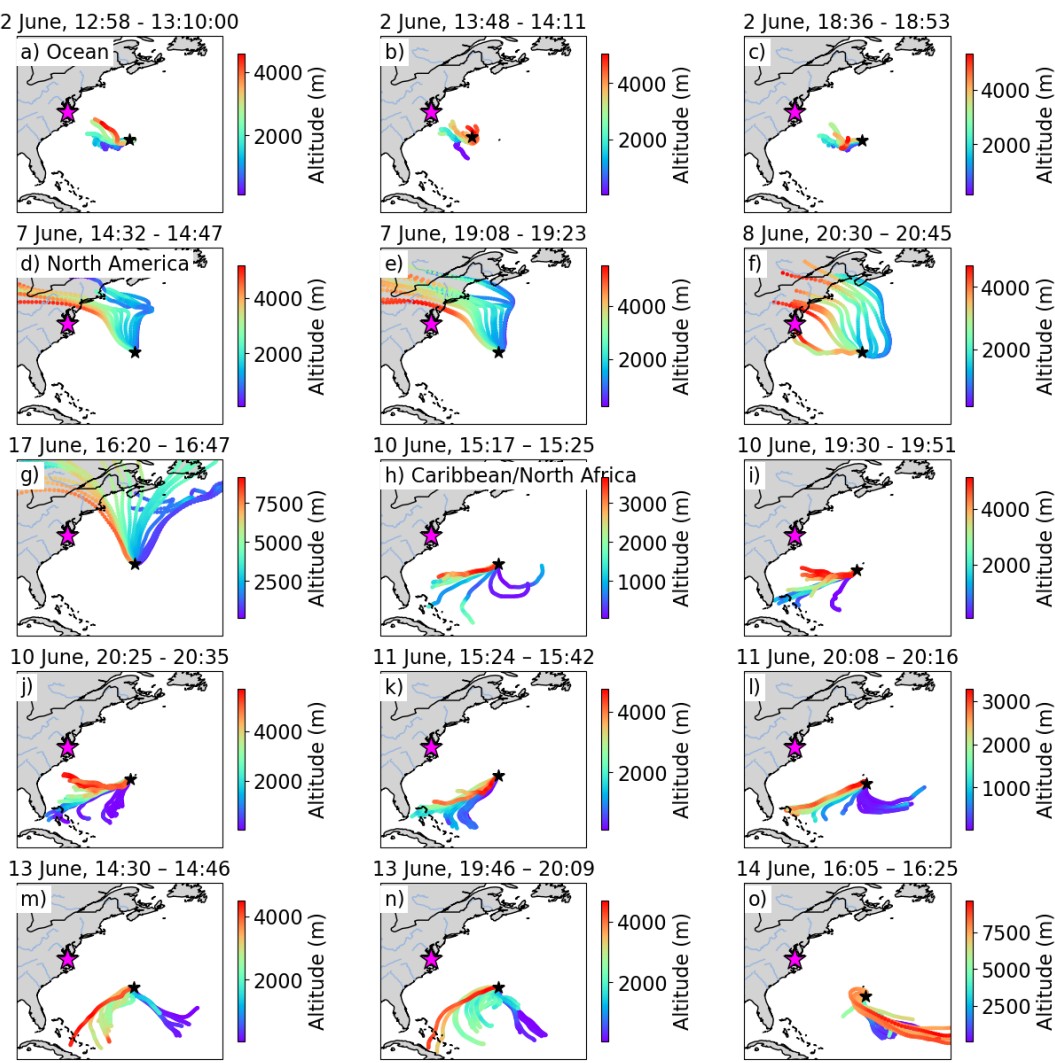


**Figure 2: HYSPLIT three day back trajectories ending at the location of the Falcon aircraft at different vertical levels during individual spiral soundings. Panels a-c are categorized as Ocean, d-g as North America, and h-o as Caribbean/North Africa. The color bar indicates the altitude at each time step of individual trajectories. The pink and black stars correspond to NASA Langley Research Center (Hampton, Virginia) and Bermuda, respectively. The date and UTC time range of spirals are above each panel.**




## 3.2 Meteorology and trace gases

We next discuss measurements pertaining to basic meteorological variables (Table 3 and Fig. 3) and trace gases (Table 3 and Fig. 4). From the start, we note that in cases of multiple spirals on the same day, there was no significant change
between morning and afternoon measurements of trace gas, meteorological, and aerosol parameters. In-situ measurements of temperature show expected reductions from the lowest kilometer (~ 18 – 21°C) to 12 – 13°C in the FT between 1 – 3.5 km. While relative humidity expectedly decreased too, the reduction for North American air in the FT (1 – 3.5 km) was more pronounced (RH ~ 22%) than for other air mass types (51 – 69 %) when compared to the lowest kilometer (73 – 82%). Soundings that could reach the highest altitudes in this study, show that there were air
mass layers aloft (>5 km) where moisture generally increased for North American and Caribbean/African air masses. Dropsonde profiles were only available for North American and Caribbean/North Africa air masses, and they expectedly are consistent with the in-situ data with the added feature shown of how the dry North American layer extends from ~1.8 - 5 km with a moister layer above. While there were sparse data passing quality control criteria for wind speed from the Falcon during the spirals, wind speed data are shown only from dropsondes (only available for
North America and Caribbean/North Africa air), revealing wind speeds typically between 5 and 10 m s$^{-1}$, with more of an increase with altitude for the North America category.

The cumulative amount of precipitation along trajectories up to the point of the Falcon spirals provide a sense of the potential for wet scavenging of particles and soluble gases to occur in transported air masses (Table 3). North American air masses exhibited the highest APT value (19.6 mm) for air sampled in the lowest km during spirals as compared to
Ocean (2.2 mm) and Caribbean/North Africa (5.4 mm) air masses (Table 3). The enhanced level for North American air is due to RF 178. APT values were generally lower in the 1-3.5 km range as compared to < 1 km.

Carbon dioxide ($CO_2$) had slightly higher values in the FT and with most variability for North American air masses (Fig. 4). Median $CO_2$ values (ppm) in the MBL/FT (1 – 3.5 km) were 418.63/420.16, 421.29/421.56 and 420.42/420.48 for North America, Ocean, and Caribbean/North Africa, respectively (Table 3).

We next examine ozone ($O_3$), which is a critical source of tropospheric OH radical and produced by sunlight, volatile organic compounds (VOC), and nitrogen oxides ($NO_x$), the latter two of which come from a plethora of anthropogenic and natural sources (Cooper et al., 2014). Ozone concentrations generally increased with altitude up to 5 km for all air mass types, with systematically higher (lower) values for North America (Caribbean/Africa) at a given altitude. Median values (ppbv) in the MBL/FT (1 – 3.5 km) were 39.60/55.50 (North America), 18.93/43.50 (Ocean), and
14.78/24.20 (Caribbean/North Africa). The maximum ozone value (102.9 ppb) was observed for the North America category at 4.8 km. Elevated $O_3$ events have been documented previously over Bermuda in the springtime due to presumed downward mixing of air from the stratosphere (Oltmans and Levy, 1992; Moody et al., 1995; Milne et al., 2000), and the results here are consistent with a vertical gradient with enhanced values aloft. One study showed levels exceeding 100 ppbv from 5 to 15 km above sea level over Bermuda in July 1993 from North American air masses that
were 2-5 days old (Merrill et al., 1996). Furthermore, Dickerson et al. (1995) noted that more than half of the air during "ozone episodes", defined as having > 40 ppbv, over Bermuda in June 1992 were due to transport from the boundary layer of eastern North America. For context, Zawadowicz et al. (2021) reported summertime $O_3$ levels of



10 – 40 ppbv in the MBL over the eastern North Atlantic, which is consistent with all air mass types except those from North America.

Vertical profiles of $CH_4$ and CO exhibited varying degrees of variation across the range of altitudes profiled (Table 3). CO is considered a "quasi-conservative" marker for anthropogenic emissions (e.g., fossil fuel and biomass combustion, oxidation of methane and non-methane hydrocarbons) (Fishman and Seiler, 1983; PéTron et al., 2002) with a lifetime of approximately ~1 – 3 months (Holloway et al., 2000). Methane on the other hand has a longer lifetime (~ 10 yrs) with diverse sources ranging from soils to fossil fuel combustion (Robson et al., 2018). There was

a consistent separation in concentrations at fixed altitudes up to 5 km between the air mass types: North America > Ocean > Caribbean/Africa. North American CO and $CH_4$ levels showed the most pronounced reduction with altitude in the lowest 5 km. The vertical structures of the trace gases exhibit relationships with the T/RH profiles, suggestive of different air mass layers; this was most distinct for North American air around 1.1 and 5 km and Ocean air around 2 km. Past work over the remote Atlantic such as over the Azores (Honrath et al., 2004) showed that elevated CO and

$O_3$ were attributed to North American pollution outflow and transported smoke plumes from source regions as far as Siberia. Typical CO values in the eastern North Atlantic during summertime were 60 – 100 ppb (Zawadowicz et al., 2021), which were lower than in winter due to more efficient removal in the summer from oxidation with the OH radical (Logan et al., 1981). Corral et al. (2021) showed that the lowest minimum values across the entire northwest Atlantic of any season were in the summer over the open ocean by Bermuda. The CO values for North American air

masses (> 90 ppb), and to a lesser extent Ocean air masses (~80 – 90 ppb), over Bermuda tend to be higher than reported background values (~ 80 ppb) devoid of urban influence (Zuidema et al., 2017; Shilling et al., 2018).

For all four trace gases, Ocean profiles exhibited the least variability at fixed altitudes (i.e., smallest whiskers in Fig. 4), likely due to the three spirals from this category being on the same day without much change across 6 hours. FT values of $O_3$ in the Ocean category show pronounced enhancements relative to MBL values, even approaching $O_3$

levels from North American air masses, suggestive that FT air in this category is probably influenced by North American emissions. Similarly, Caribbean/North African air had a steeply rising $O_3$ profile up to the highest point shown in Fig. 4 at ~8.5 km, which was comparable to North American values.

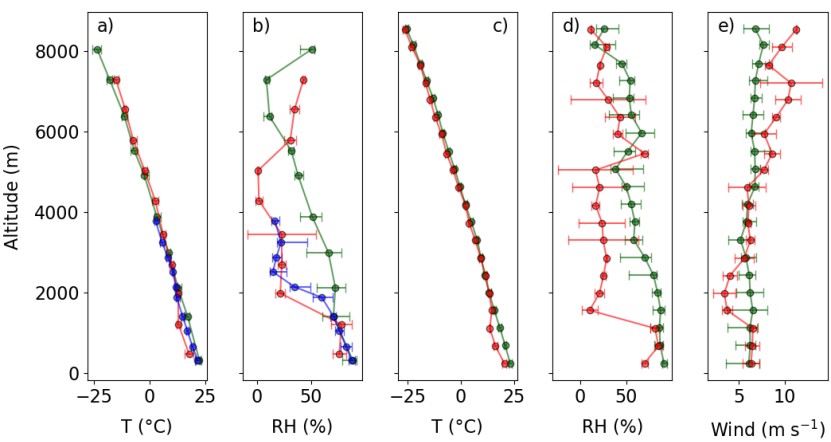



**Figure 3: Vertical distribution of meteorological variables as measured (a-b) in-situ by the Falcon and (c-e) with dropsondes launched from the King Air. Shown are temperature (T), relative humidity (RH), and wind speed (only for King Air) grouped into similar air mass source categories (red = North America; green = Caribbean/North Africa; blue = Ocean). Markers are median values and whiskers are 25th/75th percentiles. Data were unavailable for the Ocean category for dropsondes.**


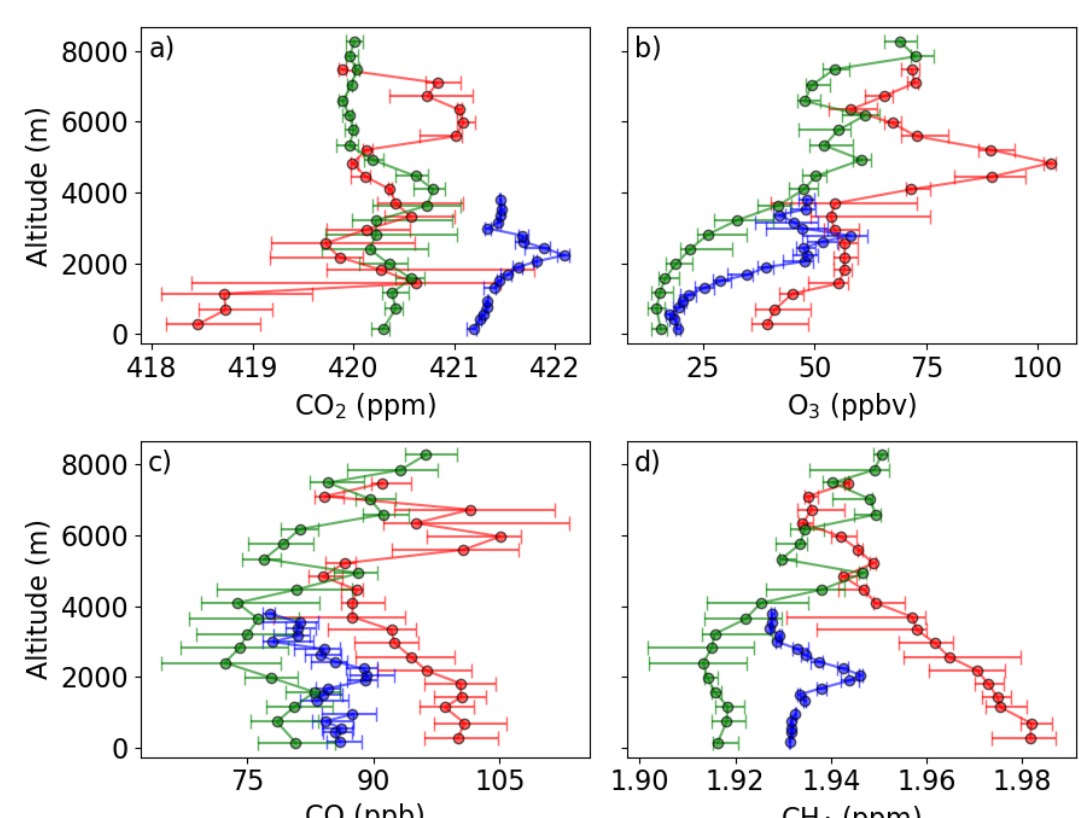

**Figure 4: Vertical distribution of trace gas concentrations for each air mass source region (red = North America; blue = Ocean; green = Caribbean/North Africa). Markers are median values and whiskers are 25th/75th percentiles. Marine**
**boundary layer heights and aerosol mixing layer heights are shown in Table 1 for reference.**

**Table 3.** Median statistics for meteorological and trace gases, as measured by the Falcon aircraft, for three source regions below 1 km and between 1 and 3.5 km. Refer to Figs. 3-4 for the full vertical profiles.

| < 1 km | T (°C) | RH (%) | APT (mm) | $CH_4$ (ppm) | $CO_2$ (ppm) | CO (ppb) | $O_3$ (ppbv) |
|---|---|---|---|---|---|---|---|
| North America | 16.98 | 78.00 | 19.60 | 1.98 | 418.63 | 100.90 | 39.60 |
| Ocean | 20.03 | 83.67 | 2.20 | 1.93 | 421.29 | 86.20 | 18.93 |
| Caribbean/North Africa | 21.11 | 83.65 | 5.40 | 1.92 | 420.42 | 80.88 | 14.78 |

1 – 3.5 km





| | | | | | | | |
|---|---|---|---|---|---|---|---|
| North America | 11.91 | 24.10 | 1.10 | 1.97 | 420.16 | 97.85 | 55.50 |
| Ocean | 11.57 | 44.70 | 1.35 | 1.93 | 421.56 | 84.80 | 43.50 |
| Caribbean/North Africa | 12.16 | 76.90 | 4.90 | 1.92 | 420.48 | 79.67 | 24.20 |


### 3.3 Particle number and volume concentrations

We next discuss particle number concentration statistics for different size ranges (Fig. 5): 3-10 nm, 10-100 nm, 0.1-1 µm, 1-5 µm, >3 µm. Particle concentrations at the smallest sizes (3-10 nm) generally increased with altitude, consistent

with literature suggesting that new particle formation in marine regions is more prevalent in the free troposphere (Bianchi et al., 2016; Clarke et al., 1998, 2013; Xiao et al., 2023) due to usually lower aerosol surface areas as sea salt in particular is a significant sink for precursor vapors in the MBL. Median number concentrations between 3-10 nm in the lowest 1 km and between 1-3.5 km ranged between $78 – 326$ cm$^{-3}$ and $135 – 266$ cm$^{-3}$, respectively, for the three air type categories. North American air generally exhibited the highest concentrations at all altitudes due likely to

higher levels of precursor vapors (e.g., $NO_x$, $SO_2$, VOCs) from continental outflow (e.g., Corral et al., 2022). These particles represent an important source of CCN due to their ability to grow into CCN-relevant sizes (e.g., Zheng et al., 2021). The ratio of number concentration above 3 nm versus 10 nm ($N_3$:$N_{10}$) is commonly used to infer about the influence of new particle formation (NPF) (Corral et al., 2022 and references therein). This ratio increases with altitude for the full duration of the spiral soundings conducted, ranging at the lowest altitude from ~1.25 to ~1.8 above 8 km.

Table 4 shows that the ratio values were similar amongst the air mass types below 1 km and between $1 – 3.5$ km. Particles in the ranges of $0.01 – 0.1$ µm and 0.1-1 µm showed comparable number concentrations for the three air categories across the vertical range of measurements, with a general reduction up to approximately 2 km followed by an increase up to higher altitudes. Because the number concentrations of different size ranges within the submicron aerosol population were higher in the FT (especially above 5 km), any air entrained from the FT into the MBL can be

an influential source of aerosol and especially CCN around Bermuda. Consistent with most of the trace gases, number concentrations were highest for North American air below a diameter of 1 µm, with median values (10-100 nm/0.1-1 µm) being 202/297 cm$^{-3}$ for < 1 km and 138/155 cm$^{-3}$ for $1 – 3.5$ km.

While much lower in number concentration than the submicron size fraction, supermicron particles are significant owing to their efficient ability to scatter light and influence clouds via effects documented for giant CCN (Houghton,

1938). Figure 5 shows a general reduction in number for particles with $D_p$ between 1 - 5 µm (from LAS) and > 3 µm (from FCDP) with altitude up to approximately 2 km, above which values were negligible; note that the FCDP data are at ambient conditions and not dried like for LAS. The vertical profiles for supermicron particle concentration are generally consistent with those reported elsewhere for air masses closer to the U.S. East Coast (Gonzalez et al., 2022) and over Bermuda (Horvath et al., 1990; Kim et al., 1990). Sources of these larger particles will be discussed in

subsequent sections, but the likely candidates expectedly are sea salt and dust particles. The Caribbean/North Africa category exhibited the highest median supermicron number concentrations between $1 – 3.5$ km (LAS: 1 cm$^{-3}$).

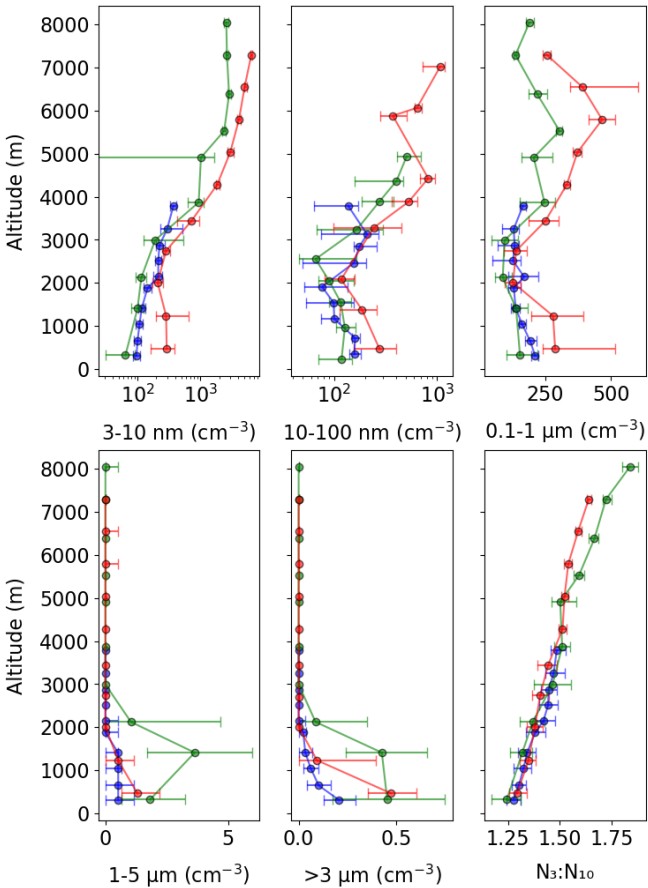

**Figure 5: Vertical distribution of aerosol number concentration (MBLHs/MLHs in Table 1 for reference) for flight data grouped into similar air mass source categories (red = North America; blue = Ocean; green = Caribbean/North Africa). Markers are median values and whiskers are 25th/75th percentiles. Note that all diameter ranges are for dry particles except for > 3 μm, which is at ambient RH conditions.**

Figure S1 and Table S2 show results for aerosol volume concentrations to complement the previous discussion, with the main conclusion that the sub-100 nm volume concentration vertical profiles follow number concentration in terms of changes with altitude and how North America exhibits highest values. For 0.1-1 μm, 1-5 μm, and >3 μm, the highest volume concentration was for Caribbean/North Africa up to 2 km, without much variation above that level between categories. These findings contradict previous studies conducted in Bermuda (Horvath et al., 1990; Kim et al., 1990) in that we did not observe a general decrease in both the number and volume concentration from the MBL to the FT. Instead, we observed the highest values for number and volume at higher altitudes, specifically for certain size ranges (i.e., below 1 μm) and types of air masses. It is noted that those earlier studies were limited in terms of what higher altitudes they could report data at, with Kim et al. (1990) and Horvath et al. (1990) reporting FT data at altitudes of 2750 m and 1500 m, respectively; additionally, those studies were limited to measurements with wing-mounted optical probes, which introduce additional uncertainties that are well documented in the literature (Porter and Clarke, 1997;





Reid et al., 2001, 2006). These comparisons with past work motivate the importance of the ACTIVATE data to give a
more complete picture of the vertical column above this remote marine location of the northwest Atlantic. Our results
are also consistent with earlier reports of how North American outflow can especially enhance small particle number
concentrations in the remote marine atmosphere of the Atlantic (Anderson et al., 1993).

**Table 4.** Summary of median aerosol number concentration statistics for different diameter size ranges for the three
major air mass types. Note that all diameter ranges are for dry particles except for > 3 μm, which is at ambient RH
conditions. The farthest right column is the ratio of number concentration above 3 nm versus 10 nm and is unitless.

| | $N \ (cm^{-3})$ | | | | | | |
|---|---|---|---|---|---|---|---|
| <1 km | 3 - 10 nm | >10 nm | 10 - 100 nm | 0.1 - 1 μm | 1 - 5 μm | > 3 μm | $N_3{:}N_{10}$ |
| North America | 326 | 1118 | 202 | 297 | 2 | 0.4 | 1.30 |
| Ocean | 99 | 334 | 167 | 194 | 1 | 0.1 | 1.29 |
| Caribbean/North Africa | 78 | 343 | 116 | 146 | 2 | 0.4 | 1.27 |
| 1 – 3.5 km | | | | | | | |
| North America | 266 | 708 | 138 | 155 | 0 | 0.0 | 1.38 |
| Ocean | 191 | 442 | 99 | 140 | 0 | 0.0 | 1.41 |
| Caribbean/North Africa | 135 | 461 | 119 | 127 | 1 | 0.2 | 1.36 |

### 3.4 Lidar retrieval results

The HSRL-2 on the higher-flying King Air provided vertically-resolved data for aerosol extinction, which provide
additional context for aerosol vertical structure (Fig. 6). As expected, the vertical profiles follow the aforementioned
number and volume concentration results for supermicron particles owing to the sensitivity of light scattering to
aerosol surface area. Extinction values exhibit peaks in the lowest 2 km for North America and Caribbean/North Africa
categories, consistent with higher RHs and size-resolved particle concentration results. Above 2 km, there is a steep
decrease. An exception is a peculiar layer of enhanced extinction around 5-6 km for the North America category due
to one particular spiral on 17 June 2022. As shown in Fig. S2, this was a case of smoke transport from the southwestern
U.S. based on HYSPLIT back-trajectory results, a spatial map of smoke over North America from the Navy Aerosol
Analysis and Prediction System (NAAPS; https://www.nrlmry.navy.mil/aerosol/) (Lynch et al., 2016), and HSRL-2
aerosol type results pertaining to this flight. Figure 4 also shows CO enhancements for North American air around 5-
6 km too. This highlights the importance of smoke transport during the summer to areas as far offshore of the U.S. as
Bermuda, as was alluded to by past work (Mardi et al., 2021). The aerosol extinction vertical distributions are generally
consistent with those reported previously over Bermuda from the Cloud-Aerosol Lidar with Orthogonal Polarization
(CALIOP) sensor on the Cloud-Aerosol Lidar and Infrared Pathfinder Satellite Observations (CALIPSO) satellite in
terms of the influence of sea salt and dust in summer months (Aldhaif et al., 2021). The Caribbean region had the
highest extinction values, reaching as high as 0.17 $km^{-1}$ at an altitude of 0.17 km (RF 176 – 14 June 2022). North
America's maximum extinction was 0.1 $km^{-1}$ at an altitude of 0.94 km (RF 167 – 7 June 2022).



Vertical profiles of lidar ratio and aerosol depolarization ratio are useful to differentiate between aerosol types (Burton et al., 2012; Groß et al., 2013). HSRL lidar ratios at 532 nm for Saharan dust tend to be around ~45-55 sr (Burton et al. 2012; Groß et al., 2013), whereas values are lower for marine aerosol (< 30 sr) and slightly higher for smoke and urban pollution aerosol (~ > 60 sr); note that the documented ranges can be broader than values reported here (e.g.,

Burton et al. 2012; Groß et al., 2013). Values for North America tended to be below 30 sr in the lowest km and in excess of 70 sr above 2 km, consistent with clean marine and polluted continental/smoke in the MBL and FT, respectively. The high values above 2 km help support the speculation above that there was a smoke layer contributing to enhanced aerosol extinction around 5-6 km on 17 June. In contrast, Caribbean/North Africa air had values starting closer to 30 sr in the lowest km and peaking only around ~55 sr at 2.5 km, with values closer to 30-35 between 4.38

– 6.17 km, which is more reflective of values associated with Saharan dust aerosol (Burton et al., 2012; Groß et al., 2013). Depolarization ratio values below 0.05 at high RH (~> 65% RH) are common for marine aerosols (e.g., Groß et al., 2013; Illingworth et al., 2015; Ferrare et al., 2023) whereas higher values up to almost 0.25 and 0.28 have been reported for dry sea salt aerosol (Kanngießer and Kahnert, 2021; Ferrare et al., 2023) and dust (Järvinen et al., 2016; Huang et al., 2023), respectively. The results show that while North American air had depolarization ratios typically

less than 0.1, the Caribbean/North Africa category had values in excess of 0.1 and up to 0.48 at 5.28 km, supportive of dust (Burton et al. 2012; Groß et al., 2013) especially since Fig. 3 showed RH values that would rule out dry sea salt.

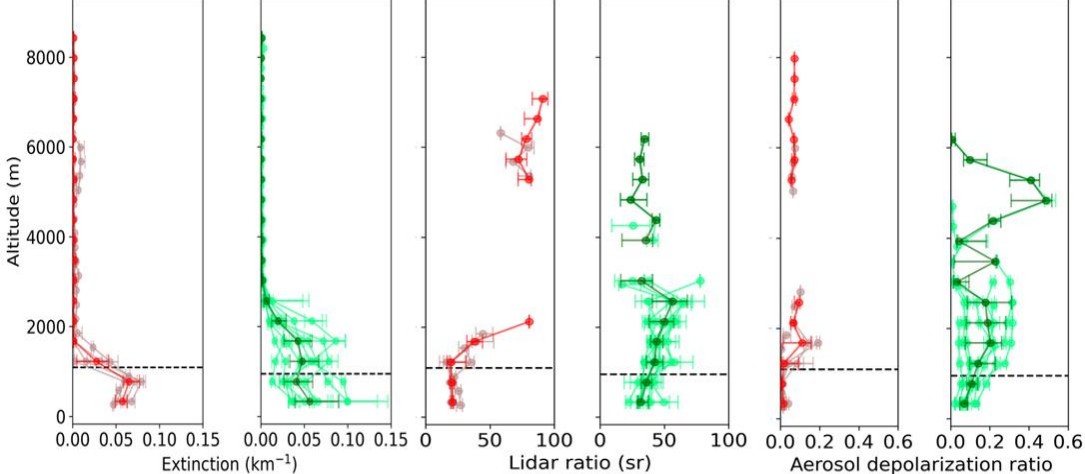

**Figure 6: Vertical distribution of (left) aerosol extinction, (middle) lidar ratio, and (right) aerosol depolarization ratio from**
**the HSRL-2 (all at 532 nm) on the King Air for (red) North America and (green) Caribbean/North Africa air categories. The darker curves represent the median and lighter curves represent data from individual overpasses. The dashed horizontal lines represent the mean aerosol MLH and markers are median values and whiskers are 25th/75th percentiles. Missing values indicate unavailable data.**

**3.5 Aerosol particle size distribution**

Detailed aerosol size distribution data are presented in Fig. 7 using both SMPS and LAS data between 3.16 to 3162 nm using two-mode log-normal curve fitting. Parameter values (number, geometric mean diameter, geometric standard



deviation) for both modes are provided in Table 5. Each air mass exhibited two distinct peaks in the BL and FT, notably in the Aitken mode and accumulation modes separated by distinct dips in number consistent with the Hoppel minimum

(Hoppel et al., 1986). This suggests that regardless of air mass source, the sampled aerosol had probably undergone some cloud processing to transfer constituents from smaller particles in the Aitken mode and gases to larger particles in the accumulation mode. This is consistent with other ACTIVATE measurements conducted around Bermuda during the time of this study with the Falcon (Crosbie et al., 2024). Expectedly, North American air exhibits systematically higher number concentration with prominent peaks at diameters of 50/143 nm and 45/106 nm for <1 km and 1-3.5

km, respectively. For Ocean, the modes were at 63/192 nm and 35/107 nm for <1 km and 1-3.5 km. Lastly, the Caribbean/North Africa category exhibited modes at 53/177 nm and 39/92 nm for <1 km and 1-3.5 km, respectively. Thus, there was a decrease in the geometric mean diameter for both modes and all air mass types as a function of altitude from < 1 km to 1-3.5 km, suggestive of the air masses not being necessarily related in the MBL and FT.


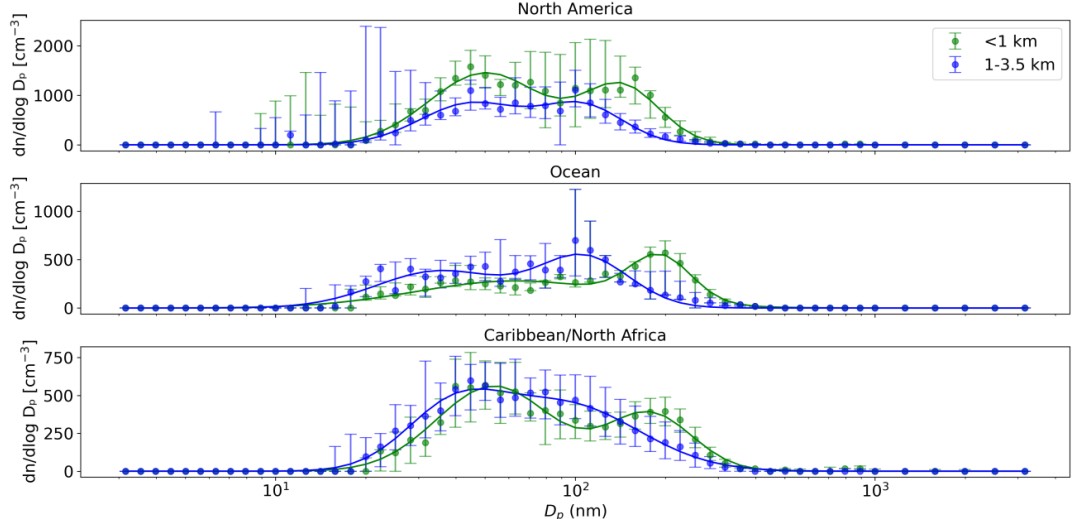

**Figure 7: Fitted aerosol number size distributions for data collected during spiral soundings categorized by the three air mass sources. Data are divided between below and above 1 km. Markers are median values and whiskers are 25th/75th**
**percentiles.**

**Table 5.** Two-mode lognormal model parameters for spirals categorized by source region. N = total number concentration; $D_g$ = geometric mean diameter, $\sigma_g$ = geometric standard deviation.

| < 1 km | $N_1$ (cm$^{-3}$) | $D_{g,1}$ (nm) | $\sigma_{g,1}$ | $N_2$ (cm$^{-3}$) | $D_{g,2}$ (nm) | $\sigma_{g,1}$ |
|---|---|---|---|---|---|---|
| North America | 1594 | 50 | 1.54 | 855 | 143 | 1.34 |
| Ocean | 538 | 63 | 2.15 | 274 | 192 | 1.27 |
| Caribbean/North Africa | 620 | 53 | 1.55 | 312 | 177 | 1.39 |
| 1 – 3.5 km | | | | | | |
| North America | 829 | 45 | 1.48 | 613 | 106 | 1.37 |
| Ocean | 473 | 36 | 1.63 | 455 | 107 | 1.41 |



| | | | | | | |
|---|---|---|---|---|---|---|
| Caribbean/North Africa | 345 | 39 | 1.45 | 627 | 92 | 1.76 |


### 3.6 Aerosol composition

Discussion next explores AMS submicron composition (Fig. 8 and Table 6). The total AMS mass concentration was highest for North America across the range of altitudes sampled, with mean values (µg m$^{-3}$) below 1 km being 2.66 (North America), 2.03 (Ocean), and 1.53 (Caribbean/North Africa). Values tended to decrease with altitude with the

exception of a notable enhancement between 5 – 7 km (peaking at 2.78 µg m$^{-3}$ observed at ~ 6500 m) for North American air in line with the enhancement in aerosol volume concentration already shown (Fig. S1) that was attributed to smoke (Fig. S2).

Speciated mass fractions for North American air were distinctly different from the other two categories owing to the dominance of organics, reaching values of 0.49 and 0.56 for < 1 km and 1-3.5 km, respectively. Sulfate was the next

most abundant constituent, followed by ammonium and with much lower mass fractions for nitrate and chloride (typically <0.03). The degree of oxygenation of the organic constituents ($f_{44}$) was the highest for North American air with mean values of 0.12 for both altitude ranges in Table 6. These general results for North American air are consistent with past work examining the first two years of ACTIVATE flights closer to the U.S. East Coast (Dadashazar et al., 2022), where sulfate and organics dominated AMS mass concentration but $f_{44}$ values are lower in these Bermuda

profiles than the latter study's sample set that showed values typically between 0.1-0.3.

For the Ocean category, sulfate was the most abundant constituent in the lowest 1 km with a mean mass fraction of 0.64, followed by ammonium (0.24), and organics (0.11). The composition of this category shifted between 1-3.5 km to have comparable levels of sulfate and organics with their mean mass fractions being 0.38 and 0.40, respectively, while ammonium was 0.20.

Lastly for the Caribbean/North Africa category, the mean mass fractions were similar to the Ocean category with a sulfate-rich profile in the lowest km (mass fraction of 0.65), that became somewhat more comparable between sulfate and organics between 1-3.5 km with mean mass fractions of 0.49 and 0.37, respectively. Extending up higher to 8 km, the organic mass fraction for this category kept increasing with values between 0.5-1.0 for various altitude bins between 2 and 6 km.

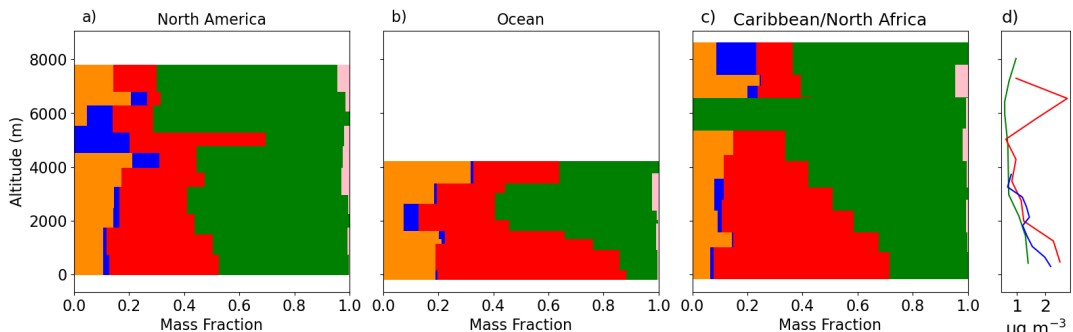


**Figure 8: Vertical distribution of AMS (a-c) speciated mass fractions (red = sulfate; green = organics; blue = nitrate; orange = ammonium; pink = chloride) and (d) total mass concentrations (total mass threshold > 0.4 µg m$^{-3}$) for each air mass type (red = North America; blue = Ocean; green = Caribbean/North Africa).**



**Table 6.** Mean statistics for AMS speciated mass fraction and total concentration (total mass threshold > 0.4 µg m⁻³), along with the $f_{44}$ ratio, for the three air mass source regions below and above 1 km. Also shown are mean f(RH) values on the far right.

| <1 km | Organics | SO₄ | NO₃ | NH₄ | Cl | $f_{44}$ | Total AMS mass (µg m⁻³) | f(RH) |
|---|---|---|---|---|---|---|---|---|
| North America | 0.49 | 0.39 | 0.02 | 0.10 | 0.00 | 0.12 | 2.66 | 1.38 |
| Ocean | 0.11 | 0.64 | <0 | 0.24 | 0.01 | <0 | 2.03 | 1.48 |
| Caribbean/North Africa | 0.19 | 0.65 | 0.01 | 0.16 | 0.00 | <0 | 1.53 | 0.95 |
| 1 – 3.5 km | | | | | | | | |
| North America | 0.56 | 0.28 | 0.01 | 0.15 | 0.01 | 0.12 | 1.41 | 1.38 |
| Ocean | 0.40 | 0.38 | 0.02 | 0.20 | 0.01 | 0.04 | 1.20 | 1.22 |
| Caribbean/North Africa | 0.37 | 0.49 | <0 | 0.15 | <0 | 0.05 | 0.99 | 0.88 |

Table 7 summarizes the water-soluble composition results from the PILS, which offers insight into species not measured by the AMS and also a look at a broader size range up to 5 µm. PILS data were combined for entire spiral soundings without separation into altitude layers owing to the lengthier time resolution to obtain samples as compared to the other instrument datasets. When viewing total mass as the sum of concentrations (µg m⁻³) of the species shown (using total sulfate rather than nss sulfate for calculation of total mass), North America exhibited the highest mean value (6.33), followed by Caribbean/North Africa (5.18) and Ocean (3.65). Noteworthy was the stretch of three flights between 7-8 June for North American air where the total masses were significantly higher than other flights (up to 20.03 µg m⁻³ for RF166), which we attribute to the weather conditions associated with the passage of tropical storm Alex where high winds could have helped promote more sea salt emissions. This is confirmed by the high Na⁺ and Cl⁻ mass fractions for those particular flights (0.29-0.55 for Na⁺; 0.32-0.47 for Cl⁻). The high APT values associated with RF178 do not translate to significantly lower PILS mass concentrations as compared to other flights; RF178 had the second highest total PILS mass (12.91 µg m⁻³) due to high Na⁺ and Cl⁻ contributions from sea salt. As it relates to APT, the mean value for each spiral showed no significant relationship with either AMS/PILS total mass nor the total volume concentration between 0.01-5 µm from the LAS (Fig. S3). Past work for surface aerosol over Bermuda showed that the winter season is characterized by most wet scavenging as compared to other seasons in terms of air masses arriving at Bermuda from distant continents (Dadashazar et al., 2021). Aside from limitations of relying on APT as a metric for wet scavenging of aerosol, a lack of any relationship with APT (Fig. S3) also builds on findings that precipitation may actually be a source of particles rather than just a sink (e.g., Khadir et al., 2023).

The unique feature from the Caribbean/North Africa spirals was the higher mass fraction for Ca²⁺, especially on RF 174 (0.31), which is supportive of the presence of dust and consistent with results shown already for how supermicron particles were enhanced for this air mass type (e.g., Figs. 5-6, and S1) for particles larger than 1 µm.



We assessed the potential for sea salt reactivity in the form of chloride depletion by comparing the mass ratio of $Cl^-$ :$Na^+$ between different spiral soundings, with a value of 1.8 being attributed to pure sea salt. The mean $Cl^-$:$Na^+$ ratio for the three air mass types was 0.70 (Ocean), 1.39 (North America), and 1.40 (Caribbean/North Africa). While the Ocean spirals exhibited the lowest overall mass concentrations, they did have the highest relative combined amount of nitrate, nss sulfate, and oxalate, which are key acidic species participating in chloride depletion. The results suggest that the sea salt sampled, regardless of air mass origin, had potentially experienced a loss of chloride due to reactions with acidic species. We caution though that the use of the $Cl^-$:$Na^+$ ratio to assess chloride depletion has limitations owing to the assumption that sea salt is the only source of the two species used in the ratio, which at least becomes a challenge for the Caribbean/North Africa category where there is evidence supporting the presence of dust. The reader is referred elsewhere for a more detailed discussion of sea salt reactivity during ACTIVATE and over Bermuda (Edwards et al., 2023).


**Table 7.** Mass fraction and total mass of water-soluble aerosol chemical composition (PILS) for the fifteen spirals categorized based on air mass source region. Mean values for each air mass category are shown in the bottom three rows.

| Research flight-date | Total Mass ($\mu g\,m^{-3}$) | $Na^+$ | $K^+$ | $Mg^{2+}$ | $Ca^{2+}$ | $Cl^-$ | $NO_3^-$ | $SO_4^{2-}$ | NSS $SO_4^{2-}$ | Oxalate | $Cl^-$:$Na^+$ | Air mass category |
|---|---|---|---|---|---|---|---|---|---|---|---|---|
| 162 - 06/02/2022 | 2.00 | 0.17 | 0.01 | 0.03 | 0.04 | 0.15 | 0.26 | 0.35 | 0.31 | 0.00 | 0.89 | Ocean |
| 162 - 06/02/2022 | 5.24 | 0.21 | 0.01 | 0.02 | 0.02 | 0.19 | 0.22 | 0.32 | 0.27 | 0.00 | 0.93 | Ocean |
| 163 - 06/02/2022 | 2.09 | 0.40 | 0.02 | 0.04 | 0.03 | 0.06 | 0.16 | 0.28 | 0.18 | 0.01 | 0.27 | Ocean |
| 166 - 06/07/2022 | 20.03 | 0.29 | 0.01 | 0.02 | 0.01 | 0.45 | 0.10 | 0.11 | 0.04 | 0.00 | 1.55 | North America |
| 167 - 06/07/2022 | 7.72 | 0.55 | 0.01 | 0.04 | 0.00 | 0.32 | 0.03 | 0.05 | 0 | 0.00 | 0.59 | North America |
| 168 - 06/08/2022 | 9.63 | 0.32 | 0.01 | 0.02 | 0.02 | 0.47 | 0.04 | 0.12 | 0.04 | 0.00 | 1.47 | North America |
| 169 - 06/10/2022 | 7.53 | 0.26 | 0.00 | 0.01 | 0.01 | 0.42 | 0.09 | 0.21 | 0.14 | 0.00 | 1.63 | Caribbean/North Africa |
| 169 - 06/10/2022 | 3.70 | 0.36 | 0.01 | 0.03 | 0.03 | 0.15 | 0.17 | 0.23 | 0.14 | 0.00 | 0.61 | Caribbean/North Africa |
| 170 - 06/10/2022 | 7.47 | 0.20 | 0.02 | 0.03 | 0.01 | 0.30 | 0.18 | 0.27 | 0.22 | 0.01 | 1.52 | Caribbean/North Africa |
| 171 - 06/11/2022 | 3.26 | 0.22 | 0.02 | 0.03 | 0.07 | 0.19 | 0.23 | 0.24 | 0.19 | 0.00 | 0.85 | Caribbean/North Africa |
| 172 - 06/11/2022 | 5.64 | 0.10 | 0.02 | 0.06 | 0.07 | 0.13 | 0.29 | 0.28 | 0.25 | 0.05 | 2.18 | Caribbean/North Africa |
| 173 - 06/13/2022 | 5.31 | 0.20 | 0.01 | 0.03 | 0.02 | 0.29 | 0.18 | 0.27 | 0.22 | 0.01 | 1.45 | Caribbean/North Africa |
| 174 - 06/13/2022 | 3.15 | 0.08 | 0.05 | 0.04 | 0.31 | 0.13 | 0.16 | 0.23 | 0.21 | 0.01 | 1.50 | Caribbean/North Africa |
| 176 - 06/14/2022 | 3.86 | 0.15 | 0.01 | 0.02 | 0.03 | 0.20 | 0.27 | 0.31 | 0.27 | 0.00 | 1.29 | Caribbean/North Africa |
| 178 - 06/17/2022 | 12.91 | 0.28 | 0.01 | 0.03 | 0.00 | 0.43 | 0.12 | 0.13 | 0.06 | 0.00 | 1.54 | North America |
| Air mass category | | | | | | | | | | | | |
| North America | 6.33 | 0.33 | 0.01 | 0.03 | 0.01 | 0.43 | 0.08 | 0.11 | 0.05 | 0.00 | 1.39 | |
| Ocean | 3.65 | 0.27 | 0.02 | 0.03 | 0.03 | 0.14 | 0.20 | 0.32 | 0.25 | 0.00 | 0.70 | |



| Caribbean/North Africa | 5.18 | 0.20 | 0.02 | 0.03 | 0.06 | 0.22 | 0.20 | 0.26 | 0.21 | 0.01 | 1.40 |
|---|---|---|---|---|---|---|---|---|---|---|---|

To put the aerosol composition results in perspective, measurements in June at the Observatory of Mount Pico in the Azores (i.e., a free tropospheric site representative of transported North American air) revealed aging exceeding 10 days with the most abundant aerosol types being carbonaceous followed by sea salt with sulfate (Cheng et al., 2022). Another study at that same site reported evidence of both North American wildfire emissions and continental outflow with the latter having more oxidized organics (O:C ratio of 0.57) even though it had 3 days of transport time from the

continent versus 7-10 days for the smoke plumes that exhibited O:C ratios around 0.45-0.48 (Schum et al., 2018). Measurements in the MBL of the Azores during the Aerosol and Cloud Experiment in the Eastern North Atlantic (ACE-ENA) campaign revealed average AMS mass loadings of 0.6 µg m$^{-3}$ in summer, with sulfate dominating (69%) followed by organics (23%), ammonium (23%), and nitrate (1%) (Zawadowicz et al., 2021). They further found that North American biomass burning reaching that site enhanced aerosol levels in the FT, along with the background

CCN budget being heavily influenced by sulfate from ocean biogenic emissions. Thus, our results are consistent with the types of air masses sampled (e.g., North American outflow, including smoke) and that sulfate was more influential in the MBL versus the FT where organics were relatively more important.

**3.7 Aerosol hygroscopicity**

Figure 9a and Table 6 present vertically-resolved results for submicron f(RH), which is an indicator of sub-saturated

hygroscopicity as it compares light scattering of particles at RHs of 80% to 20%. The f(RH) values were highest for the North America and the Ocean categories. Mean values for North America/Ocean for < 1 km and 1-3.5 km were 1.38/1.48 and 1.38/1.22, respectively. The Caribbean/North Africa category exhibited lower f(RH) values for < 1 km (0.95) and 1-3.5 km (0.88). Typically, f(RH) values are ≥1, especially for urban and marine-derived particles (e.g., Shingler et al. 2016a), but it is not uncommon to have values below 1, suggesting that particles are optically smaller

after humidification (e.g., Shingler et al., 2016b). Forthcoming work aims to examine more deeply f(RH) across all ACTIVATE flights and probe deeper into the values below unity, which can have a range of explanations.

The key driver of hygroscopicity is aerosol chemical composition. Dadashazar et al. (2022) examined the first 2 years of ACTIVATE data (2020-2021) for Falcon flights closer to the U.S. East Coast and showed that the f(RH) values for "below cloud base (BCB)" legs were inversely related to organic mass fraction (MF$_{org}$). More specifically, they

extrapolated best-fit lines comparing f(RH) to organic mass fraction to show that the characteristic f(RH) value for pure inorganic (i.e., MF$_{org}$ = 0) and organic (i.e., MF$_{org}$ = 1) aerosol were 1.39 and 1.22, respectively. Figure 9b-d illustrates the relationship between f(RH) and organic mass fraction for the three air mass types, with an inverse relationship observed for the North America and Ocean categories. Those two categories varied in terms of their scatterplots due to more data available for the lowest MF$_{org}$ values (< 0.25) for Ocean in contrast to more data for the

highest MF$_{org}$ values (> 0.75) for North America. The Caribbean/North Africa category had data across the full spectrum of MF$_{org}$, and its lack of an inverse relationship appears to be due to reduced f(RH) values at the lowest MF$_{org}$ values. This may be due to species undetected by the AMS such as dust and potentially other absorbing aerosol types that are non-hygroscopic; we caution that the f(RH) data for are submicron aerosol but still there can be non



hygroscopic aerosol types such as dust in that diameter range. This is consistent with past surface-based measurements

at Tudor Hill in Bermuda (Moody et al., 2014) showing that air masses from North Africa exhibited pronounced levels of larger absorbing aerosol types (mainly dust) that are typically less hygroscopic (e.g., Denjean et al., 2015; Kandler et al., 2018; Edwards et al., 2021) as compared to combustion-derived particle types originating from North America. Similar to Dadashazar et al. (2022), we computed f(RH) for the extremes of $MF_{org}$ for the two air categories suspected to be much less affected by refractory species undetected by the AMS (North America/Ocean): pure inorganic =

1.58/1.54; pure organic = 1.07/0.85. One hypothesis for the higher pure inorganic values here as compared to the study of Dadashazar et al. (2022) is that these data are farther offshore where there is less influence from carbonaceous components and more relative influence from inorganic species such as sulfate and sea salt; future work will examine composition-hygroscopicity relationships in greater detail for the ACTIVATE dataset.

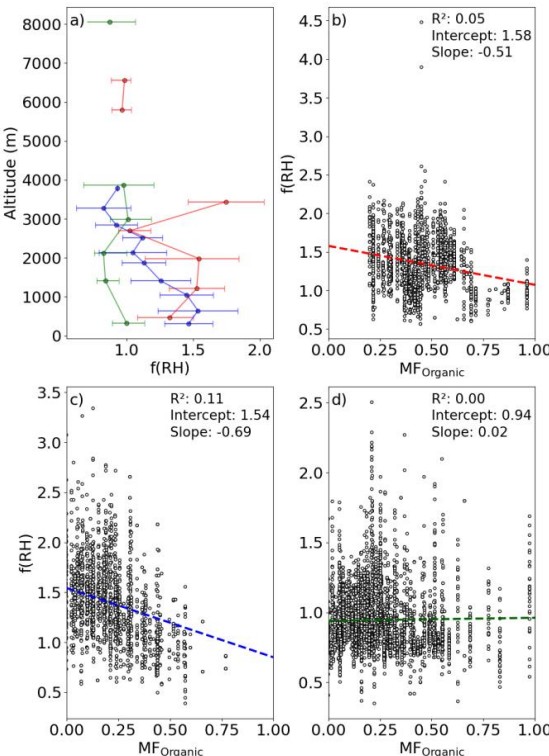


**Figure 9: (a) Vertical distribution of f(RH) values (markers = median; whiskers = 25th/75th percentiles), along with (b-d) scatterplots of f(RH) versus AMS organic mass fraction for the three air mass source regions (b = North America; c = Ocean; d = Caribbean/North Africa).**




### 4. Conclusions

A total of 15 vertical spiral soundings conducted by a HU-25 Falcon over Bermuda from 2-17 June 2022, were categorized into three air mass sources to provide insights into trace gas and aerosol characteristics. This is motivated by a large inventory of past studies (Table S1) that studied these and related topics but did not have a large extent of

vertically-resolved information, which is a gap addressed by this study. An important result in this work is that the vertical characteristics of this remote marine region depend on air mass origin, and that there is significant vertical variability in most trace gas and aerosol variables examined for a given air mass type. The latter point is important in terms of putting into perspective the limitations of column- and surface-based measurements, which have been archived for this (e.g., Aryal et al., 2014) and other remote marine regions.

Trace gases show considerable layering with altitude, with $O_3$ showing the most pronounced increase with altitude up to ~8 km with a peak value of 102.9 ppb at 4.8 km for the North American air mass category. Particle concentrations exhibit altitude-dependent variations, with new particle formation and higher submicron number concentrations more prevalent in the FT in contrast with higher supermicron number concentrations in the MBL. Contrary to previous studies in Bermuda, we did not observe a general decrease in both number and volume concentrations when ascending

from the MBL to the FT, which is presumed to be due to limitations in past works in terms of the altitude range examined in the FT. Analyses of aerosol size distribution data reveal clear peaks in the Aitken and accumulation modes, suggestive of the influence of cloud processing processes. AMS aerosol composition results show that organics dominate in North American air, whereas Ocean and Caribbean/North Africa air masses had a higher sulfate mass fraction. While sea salt and sulfate were relatively more important in the MBL, organics are relatively more influential

in the FT along with dust when trajectories came along the Caribbean/North Africa pathway. Aerosols from North American and Ocean regions had greater f(RH) values than those from the Caribbean/North Africa region, with a stronger dependence on organic mass fraction (inverse relationship) for the former two categories.

It is important to consider some limitations in this study to improve upon in future work, such as the need for more statistics (e.g., limited data here for Ocean category), a need for more detailed gas-phase measurements (e.g., $SO_2$,

DMS, volatile organic compounds), lack of measurements relating to supermicron particle impacts on aerosol optical properties, and the difficulty of having low time resolution measurements (e.g., PILS) for vertical profiles. This work motivates continued attention to the vertical inhomogeneity of trace gas and aerosol properties over remote marine regions, and provides a special data bank to help surface-based data over Bermuda where atmospheric research has been rich for several decades (Sorooshian et al., 2020).

**Data availability**

The ACTIVATE dataset is available at https://doi.org/10.5067/SUBORBITAL/ACTIVATE/DATA001 (ACTIVATE Science Team, 2020). HYSPLIT back trajectory data are available at https://www.ready.noaa.gov/HYSPLIT_traj.php and Navy Aerosol Analysis and Prediction System (NAAPS) data are available at https://www.nrlmry.navy.mil/aerosol_temp/loop_html/aer_globaer_noramer_loop_2022061800.html.




**Author contributions**

YC, EC, JPD, GSD, MAF, RAF, JWH, CAH, SK, RHM, TJS, MAS, KLT, CV, EW, and LDZ collected and/or prepared the data. TA, CS, and MRH conducted data analysis. TA and AS conducted data interpretation. TA and AS prepared the manuscript with editing from all co-authors.

**Competing interests**

At least one of the (co-)authors is a member of the editorial board of Atmospheric Chemistry and Physics.

**Disclaimer**

Publisher's note: Copernicus Publications remains neutral with regard to jurisdictional claims in published maps and institutional affiliations.


**Acknowledgements**

The authors acknowledge the pilots and aircraft maintenance personnel of NASA Langley Research Services Directorate for conducting ACTIVATE flights and all others who were involved in executing the ACTIVATE campaign. The authors gratefully acknowledge the NOAA Air Resources Laboratory (ARL) for the provision of the
HYSPLIT transport and dispersion model and READY website (https://www.ready.noaa.gov) used in this publication.

**Financial support**

ACTIVATE is a NASA Earth Venture Suborbital-3 (EVS-3) investigation funded by NASA's Earth Science Division and managed through the Earth System Science Pathfinder Program Office. University of Arizona investigators were supported by NASA grant no. 80NSSC19K0442 and ONR grant no. N00014-21-1-2115. CV and SK were funded by
DFG SPP-1294 HALO under project no. 522359172 and by the European Union's Horizon Europe program through the Single European Sky ATM Research 3 Joint Undertaking projects CONCERTO (grant no 101114785) and CICONIA (grant no 101114613).

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
