# Peer review of "Vertical variability of aerosol properties and trace gases over a remote marine region: A case study over Bermuda"

_EGUsphere, 2024_

## Author Comment (AC1)

**Manuscript ID: egusphere-2024-1065**

We express our gratitude to the reviewers for valuable feedback on the draft. We have addressed the remarks below by providing our responses in blue.

**Reviewer 1**

Ajayi et al report observations of trace gases, aerosol properties, and meteorological variables during 15 aircraft spirals near Bermuda in June of 2022. This study provides information on the vertical distribution of these observations at a location with a history of surface-based observations. This vertical information is unique for this area and will aid interpretation of surface and satellite-based observations. The paper mainly reports the observations, describing what they observed, without much detailed analysis or interpretation. However, this is a unique and relatively large data set that may be valuable for interpretation or reinterpretation of past and future studies in the Bermuda area, as well as for understanding satellite observations of trace gases and aerosol over the oceans. The vertical information was provided by 2 separate aircraft, one flying at higher altitudes and the other flying in spirals between 0.15 to 8 km altitude. The aircraft had multiple instruments to measure meteorological parameters, aerosol properties, and trace gas concentrations. The higher-flying aircraft deployed dropsondes to measure vertical profiles of temperature, relative humidity, and wind speed.

They categorize the different vertical soundings into three source regions using the HYSPLIT model. The three regions, North America, Ocean, and North Africa/Caribbean, identified by HYSLPIT are confirmed with the observed trace gases and aerosol chemical composition. They found considerable vertical variability in all three categories, with generally higher trace gas concentrations with increasing altitude (especially for ozone). Sub-micron particle concentrations also increased with altitude suggesting new particle formation in the free troposphere. Super-micron concentrations were highest near the surface and negligible above the boundary layer. Organics tended to dominate aerosol mass in the FT while sulfate and chloride was more important closer to the surface.

This paper is suitable for publication in ACP. This data could have comprised several papers that include a more detailed analysis of the observations. The data is high quality and will be useful for such future studies and thus warrants publication.

A few minor comments to improve clarity are below:

1.  Line 224: Is it 2-3 days or 2-3 hours? I think days but I'm not sure why "hours" is there.

Response: We appreciate your observation. The appropriate time frame is "2-3 days." The necessary revisions have been made to the document:

"The range of transport times from the coastline of North America to Bermuda for this category was ~2-3 days for the MBL and ~2 days for the FT."

2. The short paragraph beginning on line 272 states differences in $CO_2$ concentrations between the MBL and FT but zero analysis is given. Are these differences significant? Are they expected? Why are they different?

Response: We revised the $CO_2$ text based on this comment as we agree maybe it is misleading to get into details of variability and differences between the MBL and FT:

"Carbon dioxide ($CO_2$) levels were fairly similar at the altitudes examined (Fig. 4) with median values for the altitude bins ranging from ~418 to ~422 ppm. The most variability at fixed altitude bins, as represented by whiskers in Fig. 4, was for North American air masses."

3. Section 3.6: Remind the reader the aerosol size distribution that the AMS samples at the beginning of this section. Mean numbers are given in this section. It would be good to also state the variability in some way, such as the standard deviation of the mean.

Response: We edited Section 3.6 of the manuscript to include the size range of the AMS measurements and the standard deviation associated with mean values relevant to AMS/PILS data:

[revised manuscript text omitted]

4. Table 6 caption: "total mass threshold > 0.4…" Threshold for what? Is this the detection limit? Response: We updated and fixed the text as such in Section 2.2 and do not make mention of this mass threshold later in the paper as it is unnecessary after the first mention in Section 2.2: "To allow for better data quality, m/z 44 data are used when organic mass concentration exceeded $0.4 \ \mu g \ m^{-3}$."

5. Paragraph beginning on line 509: Please provide information on the variability about the mean for the Cl/Na ratio.

Response: We edited the text to include information about variability

"We assessed the potential for sea salt reactivity in the form of chloride depletion by comparing the mass ratio of $Cl^-$:$Na^+$ between different spiral soundings, with a value of 1.8 being attributed to pure sea salt. The mean $Cl^-$:$Na^+$ ratio for the three air mass types was $0.70 \pm 0.40$ (Ocean), $1.39 \pm 0.37$ (North America), and $1.40 \pm 0.69$ (Caribbean/North Africa)."

**Reviewer 2**

The manuscript "Vertical variability of aerosol properties and trace gases over a remote marine region: A case study over Bermuda" presents an analysis of airborne data collected during 15 vertical spiral soundings over Bermuda as part of the NASA ACTIVATE field campaign. The study focuses on understanding the vertical distribution of trace gases and aerosol properties from different air mass source regions (North America, Ocean, Caribbean/North Africa). The data from this paper is valuable to the community, as such vertical measurements of the marine atmosphere are rare.

I do not have any major concerns about this paper. There are some minor suggestions and comments:

1. Abstract: When describing the major findings, the structure of the five bullet points is inconsistent. Some are complete sentences, while others are not. I suggest using the same sentence structure throughout.

Response: We edited the structure of the major findings in the abstract of the manuscript to ensure completeness and consistency:

"(i) the strongest pollution signature is from North American air masses, while the weakest is from the Ocean category; (ii) North American air has the highest levels of CO, $CH_4$, submicron particle number concentration, AMS mass, and highest organic mass fraction along with smoke layers in free troposphere (FT); (iii) Ocean air has the highest relative amount of nitrate, non-sea-salt sulfate, and oxalate, which are key acidic species participating in chloride depletion; (iv) air masses from Caribbean/North Africa showed a pronounced coarse aerosol signature in the FT and reduced aerosol hygroscopicity, which is associated with dust transport; and (v) there is considerable vertical heterogeneity for almost all variables examined, including higher $O_3$ and submicron particle concentrations with altitude, suggestive that the FT is a potential contributor of both constituents in the marine boundary layer".

2. There are many abbreviations in this paper. I suggest summarizing them in a table for clarity.

Response: We summarized the abbreviations in the supplemental file in the form of the following table below and we introduced the table at end of introduction:

"To aid with the paper, Table S2 provides definitions of acronyms and abbreviations used subsequently."

Table S2. Summary of abbreviations and acronyms used in the paper.

| | |
|---|---|
| ACE-ENA | Aerosol and Cloud Experiment in the Eastern North Atlantic |
| ACTIVATE | Aerosol Cloud meTeorology Interactions oVer the western ATlantic Experiment |
| AMS | Aerosol Mass Spectrometer |
| APT | Accumulated Precipitation along Trajectories |
| AVAPS | Airborne Vertical Atmospheric Profiling System |
| CALIOP | Cloud-Aerosol Lidar with Orthogonal Polarization |
| CALIPSO | Cloud-Aerosol Lidar and Infrared Pathfinder Satellite Observations |
| CCN | Cloud Condensation Nuclei |
| DLH | Diode Laser Hygrometer |
| DMA | Differential Mobility Analyzer |
| DMS | Dimethylsulfide |
| FCDP | Fast Cloud Droplet Probe |
| f(RH) | Hygroscopicity Parameter |
| FT | Free Troposphere |
| GDAS | Global Data Assimilation System |
| HSRL-2 | High Spectral Resolution Lidar – Generation 2 |
| HYSPLIT | Hybrid Single-Particle Lagrangian Integrated Trajectory model |
| LaRC | NASA Langley Research Center |
| LAS | Laser Aerosol Spectrometer |
| LWC | Liquid Water Content |
| MBLH | Marine Boundary Layer Height |
| $MF_{org}$ | Mass Fraction of Organics From AMS |
| MLH | Mixed Layer Height |
| NASA | National Aeronautics and Space Administration |
| NCAR | National Center for Atmospheric Research |
| NOAA | National Oceanic and Atmospheric Administration |
| NSS | Non-Sea salt |
| PILS | Particle into Liquid Sampler |
| RF | Research Flight |
| RH | Relative Humidity |
| SMPS | Scanning Mobility Particle Sizer |
| UTC | Coordinated Universal Time |
| VOC | Volatile Organic Compounds |

Based on Figure 3, the wind speed at ~8000 m is only 5-10 m/s. Why is the wind speed so low? Additionally, I recommend not using red and green to distinguish different markers as they can be difficult to differentiate for colorblind readers.

Response: The low wind speeds of 5-10 m/s at approximately 8000 meters were checked and confirmed to be correct. They can be attributed to a host of conditions that are outside the scope of this work as we aren't focused on wind speed variations with altitude.

Also, the colors have been adjusted for colorblind accessibility (orange, blue, and green). An example is shown here for Figure 3:

[Figure]

**Figure 3: Vertical distribution of meteorological variables as measured (a-b) in-situ by the Falcon and (c-e) with dropsondes launched from the King Air. Shown are temperature (T), relative humidity (RH), and wind speed (only for King Air) grouped into similar air mass source categories (orange = North America; green = Caribbean/North Africa; blue = Ocean). Markers are median values and whiskers are 25th/75th percentiles. Data were unavailable for the Ocean category for dropsondes.**

3. Figure 5: What is the type of aerosol diameter used in Figure 5? Is it electrical mobility diameter or aerodynamic diameter?

Response: The aerosol diameters used in Figure 5 are based on optical techniques (e.g., LAS and FCDP) and not those mentioned in the comment. We feel our explanation of diameters in Section 2.2 was sufficient and no extra text is warranted in figure captions about this.

4. Figure 8: I suggest adding a legend to the figure to explain the color representations.

Response: We updated Figure 8 to include the legend in the manuscript as shown below.

[Figure]

**Figure 8: Vertical distribution of AMS (a-c) speciated mass fractions (red = sulfate; green = organics; blue = nitrate; orange = ammonium; pink = chloride) and (d) total mass concentrations for each air mass type (orange = North America; blue = Ocean; green = Caribbean/North Africa).**

5. Why does the sulfate concentration decrease with increasing altitude?

Response: The primary source of sulfate is likely the marine boundary layer and the free troposphere did not have a strong influence on sulfate levels. We did not feel the need to make changes to the manuscript in response to this comment.

6. Figure 9: How do you calculate MForangic? Seaspray contains a lot of NaCl, which is a refractory material to the AMS. I suggest using AMS organic mass/PM1 to get MForangic.

Response: The mass fraction (e.g., MF$_{organic}$) was determined by dividing the mass of each non-refractory species found by the AMS by the total mass of non-refractory species detected. Since AMS instruments generally detect particles within the size range of 0.06 – 0.6 micrometers, they cannot adequately characterize sea spray particles, especially also because they are comprised of refractory substances that the AMS struggles with. So in a way, we did quantify already AMS organic mass divided by PM$_1$ since the AMS is measuring close to what can be considered PM$_1$. As a result, we don't think changes are needed in the paper to address this comment.